# Human Uncertainty-Aware Data Selection and Automatic Labeling in Visual Question Answering

**Jian Lan**[1,2,*], **Zhicheng Liu**[1], **Udo Schlegel**[1,2], **Raoyuan Zhao**[1,2], **Yihong Liu**[1,2],
**Hinrich Schütze**[1,2], **Michael A. Hedderich**[1,2], **Thomas Seidl**[1,2]
[1]University of Munich (LMU)
[2]Munich Center of Machine Learning
* corresponding author
lan@dbs.ifi.lmu.de

## Abstract

Large vision-language models (VLMs) achieve strong performance in Visual Question Answering but still rely heavily on supervised fine-tuning (SFT) with massive labeled datasets, which is costly due to human annotations. Crucially, real-world datasets often exhibit *human uncertainty* (HU) – variation in human confidence across annotations, but standard SFT simply optimizes toward the most frequent label, disregarding HU distributions. This leaves two open questions: *How does HU affect SFT*, and *how can HU be effectively leveraged in training?* In this work, we first conduct the systematic evaluation of VLMs across varying HU levels. We have two key findings: (i) surprisingly, high-HU samples contribute little, or even degrade, model performance, and (ii) naively training on the full dataset yields under-calibrated models that fail to capture HU distributions. Motivated by these findings, we introduce **HaDola**, a **h**uman uncertainty-**a**ware **d**ata selection and aut**o**matic **la**beling framework. HaDola operates in four stages: **discriminate**, **self-annotate**, **error trigger**, and **training**, to iteratively identify harmful samples, prioritize informative ones, and bootstrap from a small seed set (5% of data). Our approach substantially reduces reliance on costly HU annotations and makes VLMs more accurate and better calibrated. Extensive experiments on VQAv2 and VizWiz datasets demonstrate that HaDola consistently matches or outperforms state-of-the-art baselines, with less training data. Our work highlights the importance of explicitly modeling HU in SFT, suggesting better utilization of HU is more effective than merely scaling up dataset size. Code: https://github.com/emsuno/hadola

## 1 Introduction

Large vision-language models (VLMs) (Wang et al., 2023; Liu et al., 2024b; Wang et al., 2024b; Bai et al., 2025) have achieved impressive progress on multi-modal tasks, with *Visual Question Answering* (VQA) (Goyal et al., 2017) being a critical benchmark for evaluating vision-language understanding and generation Zhang et al. (2025); Lan et al. (2026a; 2025b). Despite their strong performance, VLMs trained with the prevailing strategy of supervised fine-tuning (SFT) face two major limitations. First, they rely heavily on massive human-annotated data and simply scale up the dataset size without investigating how individual samples contribute to training. Some studies have explored sample difficulty in VQA (Karamcheti et al., 2021; Tan & Bansal, 2019), but they still solely depend on SFT and offer no effective solutions to data selection or reducing annotation costs. Secondly, for each sample, current VLMs optimize only for the most frequent answer label while ignoring alternative answers and, crucially, overlooking human confidence distributions (Wang et al., 2024a; Liu et al., 2024a; Wang et al., 2023). Human confidence, also known as **human uncertainty** (HU) (Lan et al., 2025a), refers to the fact that, for a sample, different annotators may provide diverse answers with varying confidence levels. HU has been demonstrated as an important and non-negligible factor for VQA in their work. As shown in Figure 1, given a question-image pair input, different humans have different answers, and even for humans providing the same answer, their uncertainty label can be

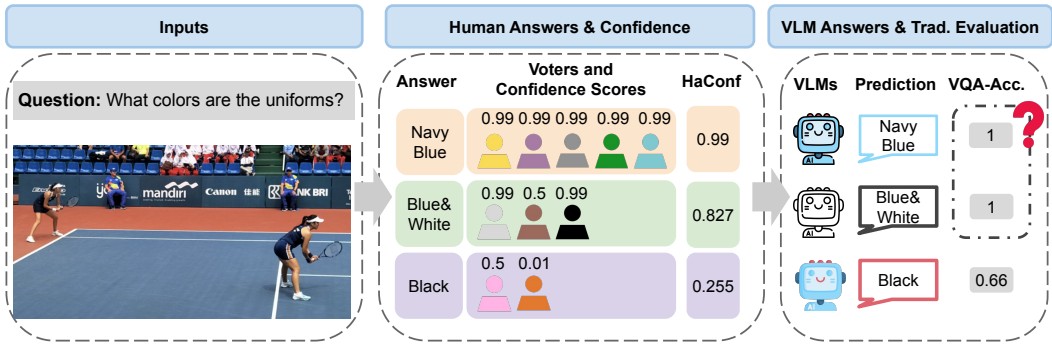

Figure 1: Illustration of human uncertainty in VQA. Different annotators may provide different answers with varying confidence levels. The HaConf Score is the average score of all annotators for each answer 1. On the right, it shows the VQA-Accuracy 4.1 of different model generations. The metric fails to reflect the HU difference, and leaves an open question: Is that an accurate metric to evaluate VLMs?

different. Neglecting HU can leave models mis-calibrated (Guo et al., 2017; Baan et al., 2022) and drive indiscriminate training data expansion, instead of exploiting HU for more effective training strategies.

To explore how HU can be appropriately incorporated in training, this work raises the following research questions: **RQ1:** To what extent does HU influence the SFT in VLMs, and what are the beneficial or harmful samples? **RQ2:** How can HU be incorporated into the training process to achieve high accuracy and better calibration? **RQ3:** Given the prohibitive cost of large-scale HU annotations, how to design efficient strategies that leverage only a small portion of HU-labeled data while maintaining strong performance?

To answer these questions, we first conduct a comprehensive evaluation of VLMs on datasets novelly stratified by different HU levels. Surprisingly, our analysis shows that high-HU samples contribute little or even harm SFT, and indiscriminate training on the full dataset yields under-calibrated models that fail to capture real-world HU distributions. These findings highlight the role of HU: it can serve as a valuable signal for guiding data selection and training strategies. Building on this insight, we introduce **HaDola** (**h**uman uncertainty-**a**ware **d**ata selection and aut**o**matic **la**beling), a novel framework that explicitly integrates HU into VLM training. HaDola follows a four-stage pipeline with advantages: (1) *discriminate*, to identify harmful versus beneficial samples; (2) *self-annotate*, to automatically refine annotations guided by HU; (3) *error trigger*, to detect and correct potential error accumulation; and (4) *training*, to fine-tune VLMs with our novel loss function. Starting from a small HU-annotated seed set (around 5% of all data), HaDola iteratively expands training supervision in a self-evolving manner, substantially reducing reliance on costly HU annotations.

Moreover, we find that the widely used evaluation metric VQA-Accuracy (Antol et al., 2015), being purely frequency-based, overlooks HU by treating all answers equally. To better align with HU, we propose a complementary measure, *HU-acc*, which weights predictions by human confidence and reveals that low- and medium-HU samples provide more effective supervision.

We conduct extensive experiments on two widely used benchmarks, VQAv2 (Goyal et al., 2017) and VizWiz (Gurari et al., 2018), with recent state-of-the-art (SOTA) VLMs including LLaVA (Liu et al., 2024b), Qwen-VL (Bai et al., 2025), InternVL (Wang et al., 2024b), and BEiT3 (Wang et al., 2023). Results show that HaDola consistently outperforms strong baselines, achieving both higher accuracy and more reliable calibration, while requiring substantially less annotated data.

**Our Contributions are:** (1) For VQA, we provide the first systematic study of how HU affects SFT in VLMs, identifying the essential role of HU as both a source of harm and a useful training signal. (2) We propose HaDola, a human-uncertainty-aware generalizable and data-efficient framework that integrates data selection and automatic labeling into VLM training, requiring only a small portion of HU annotations. We validate HaDola on VQAv2 and VizWiz with multiple SOTA VLMs, demonstrating that it achieves superior accuracy and calibration compared to baselines. (3) We

identify the limitation of the standard majority vote-based VQA-Accuracy metric, which ignores HU, and for the first time, advocate for incorporating the non-negligible HU into evaluation protocols to better assess VQA in real-world settings.

## 2 RELATED WORK

**VQA and Human Uncertainty.** VQA was first introduced by Antol et al. (2015), and follow-up work (Tan & Bansal, 2019; Li et al., 2022; Bao et al., 2022; Wang et al., 2023) has mainly optimized the standard VQA-Accuracy metric. BEiT3 Wang et al. (2023) remains the SOTA task-specific model but lacks zero-shot or generative ability. Recent VLMs (Bai et al., 2025; Liu et al., 2024b; Wang et al., 2024b) rival or surpass BEiT3, yet still rely primarily on supervised fine-tuning (SFT) without explicitly modeling HU. The non-negligible HU has only recently received research attention. Lan et al. (2025a) introduce the HUD score to quantify HU in VQAv2 with BEiT3, but leave open how HU can be exploited in training and whether it benefits beyond accuracy. Only a few datasets explicitly provide HU annotations. For example, VQA 2.0 (Goyal et al., 2017) and VizWiz (Gurari et al., 2018) include both answers and uncertainty labels from ten annotators ("How certain are you: yes, maybe, no?"). In contrast, other common datasets such as OK-VQA (Marino et al., 2019) and GQA (Hudson & Manning, 2019) do not provide HU labels. Yet prior studies (Goyal et al., 2017; Gurari et al., 2018; Tan & Bansal, 2019; Wang et al., 2023) have largely ignored this information, training models only on the majority label to maximize VQA-Accuracy. More recent work shifts attention to model uncertainty calibration across multiple answers (Yang et al., 2024; Xiong et al., 2024; Lan et al., 2026b). However, a key challenge remains: HU is rarely available at scale, making it difficult to assess whether model predictions faithfully reflect human uncertainty. This motivates the need for methods that can leverage HU more effectively and, ideally, reduce dependence on costly human annotations.

**Sample-aware training in VQA.** Recent work has examined what data and how much data to use for training. Karamcheti et al. (2021) study this via active learning (AL) (Lowell et al., 2019), showing that difficult samples cause AL to fail without offering ways to identify or handle them. Tan & Bansal (2019) measure sample difficulty by annotator label diversity, but only analyze label frequency and ignore HU. Lan et al. (2025a) are the first to explore HU in VQA, evaluating BEiT3 Wang et al. (2023) but leaving generative VLMs and HU's role in training under-explored. To reduce costly VQA annotation, Sun et al. (2024) augment medical-domain data with DPO (Rafailov et al., 2023), but assume perfect training distributions and overlook HU. Other efforts, such as domain generalization with customized loss (Huang et al., 2025), semi-supervised learning for video-QA (Mitra & Soundararajan, 2024), and selective prediction for reliability (Dancette et al., 2023; Whitehead et al., 2022) likewise aim to reduce annotation cost or improve robustness. While these approaches improve efficiency and robustness, they typically assume perfectly reliable data and do not investigate the potential benefits or harms of training on samples with high HU. This leaves open questions of how to systematically select and utilize HU-aware samples to improve both model accuracy and calibration.

## 3 BACKGROUND

**Measuring Human Uncertainty.** In a training sample, image-question-answer triplet, $s = (i, q, \mathcal{A})$, the $(i, q)$ is an image-question input pair, and $\mathcal{A}$ is an answer label set with 10 independent humans' annotations. In each human's annotation $h_n$, one annotator gives their answer $a_n$ to $q$ and also a confidence level $c_n$ to indicate whether this annotator is confident that $a_n$ is correct. $c_n$ belongs to a pre-defined category ['$yes$', '$no$', '$maybe$']. To quantify $c_n$, the latest study Lan et al. (2025a) assigns different confidence scores to $c_n$, mapping 'yes' to 0.99, 'no' to 0.01, and 'maybe' to 0.5: $\mathcal{A} = \{(a_n, c_n)\}_{n=1}^{10}, c_n \in [0.99, 0.5, 0.01]\}$. Within $\mathcal{A}$, annotators who provide the same answer $a'_m$ may assign different confidence scores $c_n$. We denote the average of these scores as $c'_m$, the **h**uman **av**erage **conf**idence of $a'_m$ (abbreviated as **HaConf**, not a new metric but a shorthand for clarity). Building on this, Lan et al. (2025a) introduce HUD (**h**uman **u**ncertainty across **d**isagreement), which aggregates HaConf scores across all different answers of a sample to quantify its HU degree. HUD is the most recent measure for capturing sample-level HU. The above are denoted as (note that HaConf

is answer-based and HUD is sample-based):

$$\text{HaConf}(a'_m) : c'_m = \frac{1}{|\{n \mid a_n = a'_m\}|} \sum_{n:a_n=a'_m} c_n, \qquad \text{HUD}(s) = \frac{1}{m} \sum_{i=1}^{m} c'_i. \tag{1}$$

**Distributions and Calibrating towards Humans.** For a model $\mathcal{M}$, the prediction distribution $\mathcal{M}(s)$ corresponds to the probabilities $P_{\mathcal{M}}(s)(Y = y|X = s)$ the model assigns to each class $y$. In VLMs, $P_{\mathcal{M}}$ is derived from the last hidden state of the neural network over the well-known logits, denoted as $L(s) = [l_1, l_2, ..., l_m]$, with $m$ as in Eq. 1. $P_{\mathcal{M}}$ is represented by applying the standard *Softmax* normalized function to $L(s)$: $P_{\mathcal{M}}(s) = Softmax(L(s))$. The human distribution $H(s) = [c'_1, \ldots, c'_m]$ is defined by the HaConf scores across different answers (Eq. 1). The standard way to assess alignment between human and model distributions is the KL-Divergence (KL) (Kullback & Leibler, 1951). Given a human distribution $H(s)$ and model distribution $P_{\mathcal{M}}(s)$, the KL score measures how much $H(s)$ deviates from $P_{\mathcal{M}}(s)$:

$$D_{\text{KL}}(H(s) \parallel P_{\mathcal{M}}(s)) = \sum_{i=1}^{m} H(s) \log \frac{H(s)}{P_{\mathcal{M}}(s)}. \tag{2}$$

# 4 HUMAN UNCERTAINTY-AWARE DATA SELECTION AND AUTOMATIC LABELING

Studying different HU levels requires dividing a dataset into varying sets. We first introduce our novel split and evaluation based on HUD and HaConf, and then introduce HaDola.

## 4.1 SAMPLES WITH HU LEVELS AND HU-BASED ACCURACY

Following prior work (Tan & Bansal, 2019; Lan et al., 2025a), we adopt a *three-level* categorization of HU (low, medium, high). In (Lan et al., 2025a), datasets are partitioned into three equal subsets by HUD scores, but this does not reflect reality, where low-HU samples dominate. Therefore, unlike their work, we divide HU by evenly splitting HUD intervals: [0.01,0.33] (high), (0.33,0.66) (medium), and [0.66,0.99] (low). As shown in Figure 2(a), our splits align better with samples' HaConf upper bound, yielding smaller variance and more balanced means. By contrast, prior splits blur medium and high sets, inflate the low set, and produce higher variance. Our design therefore achieves a sharper and more realistic distinction across HU levels.

Moreover, measuring VQA model accuracy is challenging. The standard metric VQA-acc (Antol et al., 2015) is defined as $\text{VQA-Acc}(a) = \min\left\{\frac{\#\text{humans that said } a}{3}, 1\right\}$, where only the majority vote is considered for a model-generated answer $a$. However, it ignores HU: even when multiple annotators agree, their confidence may remain low, yet the sample is still rewarded with a high score. To address this, we propose HU-acc, a HU-weighted variant defined as **HU-acc**$(a) = \text{HaConf}(a) \times \text{VQA-Acc}(a)$, which incorporates both label frequency and HU. As shown in Figure 2(b), HU-acc yields a clearer separation across subsets: training on low-, medium-, and high-HU data leads to steadily lower accuracy as HU increases. This reinforces that high-HU samples harm the SFT gained on the other two sets. Combining low- and medium-HU samples further improves performance, while adding high-HU data again impairs training, confirming their limited reliability. In contrast, VQA-acc shows only marginal differences and fails to reveal the harmful effect of high-HU samples. This discrepancy highlights the insufficiency of frequency-based evaluation and motivates the need for HU-sensitive measures. Accordingly, we replace the conventional VQA-acc supervision signal with HU-acc during SFT to enable a fairer and more informative comparison. These HU-based splits and metrics form the foundation for introducing HaDola, our HU-aware training framework.

## 4.2 HADOLA PIPELINE

**Overview.** HaDola is a model-agnostic and generalizable framework that uses HU to reduce annotation cost while matching or surpassing SOTA performance and simultaneously improving calibration. The framework operates in four key stages –*discriminate*, *self-annotate*, *error-trigger*, and *training* – as summarized in Algorithm 1 and illustrated in Figure 7. In the following, we describe each stage in detail, starting from the initialization of the HaDola.

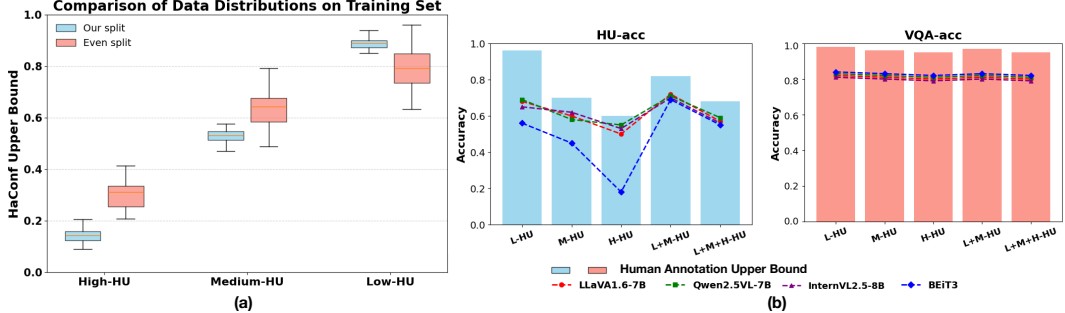

Figure 2: Results on VQAv2, (a): comparison of data distribution with different set splits design. (b): Effects of training samples with different HU degrees, and comparison with VQA-acc. The L, M, H stands for low, medium, and high. We downsample the L and M subsets to match the sample size of H. VizWiz results reach consistent findings and are in Appendix A.

**Initialization.** To capture the inherent and non-negligible HU in data, HaDola requires a small *seed set* with human annotations as an anchor for confidence distributions. Let $S_0$ be a small labeled seed set (randomly selected 5% of data from the entire dataset for human annotation), $M_{\text{init}}$ an initial VLM without SFT, and $S_r$ the remaining unlabeled data. HaDola first fine-tunes $M_{\text{init}}$ on $S_0$ to obtain $M_{\text{hu}}$, which is duplicated: one copy fixed as the *HU reference model*, the other used as the initial model for the subsequent iterative training.

**Discriminate.** HaDola aims to identify and exclude high-HU samples in each step $t$, since they are assumed to contribute neither to learning nor efficiency, which instead consumes additional human and computational resources. HaDola calculates the average KL scores between $M_{\text{hu}}$ and humans on $S_0$:

$$\tau_1 = \mathbb{E}_{a \in l_{S_0}}[D_{\text{KL}}(H(a) \,\|\, M_{\text{hu}}(a))], \ \tau_2 = \mathbb{E}_{a \in m_{S_0}}[D_{\text{KL}}(H(a) \,\|\, M_{\text{hu}}(a))], \ h_\omega = \mathbb{E}_{a \in l_{s_0} \cup m_{s_0}}[M_{\text{hu}}(a)]. \tag{3}$$

In Eq. 3, $l_{S_0}$ and $m_{S_0}$ are low- and medium-HU subsets of $S_0$. $\tau_1$ and $\tau_2$ are their average KL scores between the human reference model and true HU distribution, with $\tau_1 < \tau_2$. $h_\omega$ is the mean human confidence distribution over $l_{S_0} \cup m_{S_0}$. For each unlabeled sample $u \in S_r$ selected in this step $t$ (1% of $S_r$), HaDola computes its KL score $kl_u$ between current round $M_t$ and $h_\omega$: $kl_u = D_{\text{KL}}(h_\omega \,\|\, M_t(u))$, and retains only those with $kl_u \in [\tau_1 - \sigma, \ \tau_2 + \sigma]$. Samples outside this interval are considered high-HU or outliers and discarded, where $\sigma$ is the standard deviation of all KL scores in Eq. 3.

**Self-Annotate.** After the unhelpful instances are discarded in the *discriminate* stage, HaDola annotates each remaining instance $u$ by leveraging the last round model $M_{t-1}$: $\hat{y}_u = M_{t-1}(u)$, where $\hat{y}_u$ is the response of $M_{t-1}$ when answering $u$. It then constructs the pseudo training pair $(u, \hat{y}_u)$.

**Error Trigger.** To avoid the potential accumulation of errors in pseudo training pair $(u, \hat{y}_u)$, we introduce an error trigger mechanism, where we use gradient consistency (Mirzadeh et al., 2020) and TracIn-mini (Pruthi et al., 2020), where the former ensures that the gradient direction yielded by $(u, \hat{y}_u)$ is consistent with human-labeled data with the current model parameters, thus serving as a theoretical criterion to filter out pseudo labels that conflict with reliable supervision; the latter traces the influence of $(u, \hat{y}_u)$ across checkpoints to quantify its contribution to the model, enabling us to down-weight or discard harmful pseudo pairs. Together, these two components provide a principled mechanism to safeguard against error accumulation in self-annotated training. More theoretical derivations are in Appendix B.2. For a pseudo-labeled sample $(u, \hat{y}_u)$ at the $t^{th}$ round with model parameters $\theta_t$, we compute its gradient and the reference gradient, as the average gradient over the human-labeled seed set $S_0$ as:

$$g(u, \hat{y}_u; \theta_t) = \nabla_{\theta_t} \ell(f_{\theta_t}(u), \hat{y}_u), \qquad g_{\text{ref}}(\theta_t) = \mathbb{E}_{(x,y) \sim S_0}[\nabla_{\theta_t} \ell(f_{\theta_t}(x), y)]. \tag{4}$$

---

**Algorithm 1:** HaDola Pipeline.

---

**Input:** Seed set $S_0$ (5% labeled), unlabeled set $S_r$, initial model $M_{\text{init}}$, total rounds $T$.
**Output:** Final model $M_T$.

---

1 **Initialization:** SFT $M_{\text{init}}$ on $S_0$ to obtain $M_{\text{HU}}$; duplicate it as reference $M_{\text{HU}}$ and training copy $M_0$. Compute thresholds $\tau_1, \tau_2, h_\omega$ once on $S_0$ using $M_{\text{HU}}$ (Eq. 3).
2 **for** $t = 1$ **to** $T$ **do**
3     **Discriminate:** For each $u \in S_r$, compute $kl_u$ under $M_{t-1}$; retain $u$ if $kl_u \in [\tau_1 - \sigma, \ \tau_2 + \sigma]$.
4     **Self-Annotate:** For each $u$, assign pseudo-label $\hat{y}_u = M_{t-1}(u)$ and form $(u, \hat{y}_u)$.
5     **Error Trigger:** For $(u, \hat{y}_u)$, compute $s_g$ and $s_{\text{tracin}}$; keep if $s_g \geq \tau_g \wedge s_{\text{tracin}} \leq \tau_t$.
6     **Training:** Fine-tune $M_{t-1}$ on retained samples from all rounds with $\mathcal{L}_{\text{HaDola}}$ (Eq. 8); update $M_t$.
7 **return** $M_T$.

---

The gradient consistency score is then measured by cosine similarity:

$$s_g(u, \hat{y}_u; \theta_t) = \frac{\langle g(u, \hat{y}_u; \theta_t), \ g_{\text{ref}}(\theta_t) \rangle}{\|g(u, \hat{y}_u; \theta_t)\| \ \|g_{\text{ref}}(\theta_t)\|}. \tag{5}$$

In addition, to capture the global effect of training, we employ a simplified TracIn estimator that only requires the initial model $\theta_0$ and the current model $\theta_t$:

$$s_{\text{tracin}}(u, \hat{y}_u) \approx \langle g(u, \hat{y}_u; \theta_0), \nabla_\theta L_{\text{val}}(\theta_0) \rangle + \langle g(u, \hat{y}_u; \theta_t), \nabla_\theta L_{\text{val}}(\theta_t) \rangle, \tag{6}$$

where $L_{\text{val}}(\theta)$ is the validation loss. A pseudo-labeled sample is retained only if it passes both criteria:

$$\text{keep}(u, \hat{y}_u) = \mathbb{I}[\, s_g(u, \hat{y}_u; \theta_t) \geq \tau_g \ \wedge \ s_{\text{tracin}}(u, \hat{y}_u) \leq \tau_t \,], \tag{7}$$

where thresholds $\tau_g$ and $\tau_t$ are calculated based on low-/medium-HU subsets. We leave the details of the selection of $\tau_g$ and $\tau_t$ in Appendix B.3.

**Training.** We design a training objective that jointly pursues predictive accuracy and uncertainty calibration by leveraging a human-uncertainty (HU) reference model. The overall loss consists of three terms:

$$\begin{aligned}
\mathcal{L}_{\text{HaDola}} &= \mathbb{E}[\text{CE}(y, M_\theta)] + \beta\, \Phi + \lambda\Big(D_{\text{KL}}\big(H \parallel M_\theta\big) - D_{\text{KL}}\big(H \parallel M_{\text{HU}}\big)\Big), \\
\Phi &= D_{\text{KL}}\big(M_{\text{HU}}(\cdot \mid x) \parallel M_\theta(\cdot \mid x)\big).
\end{aligned} \tag{8}$$

The first term is the standard cross-entropy ensuring $M_\theta$ predicts labels correctly. The second term regularizes $M_\theta$ against the HU reference $M_{\text{HU}}$, preventing it from drifting too far from the HU-informed baseline. The final term compares alignment of $M_\theta$ and $M_{\text{HU}}$ with human distribution $H$, encouraging $M_\theta$ to better approximate human uncertainty and thus improve calibration.

## 5 EXPERIMENTS

### 5.1 SETUP DETAILS

**Datasets and set split.** We use two open-sourced VQA datasets with human confidence annotations: VQAv2 (Goyal et al., 2017) and VizWiz (Gurari et al., 2018). VQAv2 contains 443,757 training and 213,954 validation samples based on MSCOCO images (Lin et al., 2014), and is a widely used benchmark. VizWiz has 20,523 training and 4,319 validation samples, collected from blind users via mobile photos (e.g., asking for expiration dates). Unlike VQAv2, VizWiz poses greater challenges due to blurred or unconventional images, though entirely unanswerable ones are filtered out (details in Appendix C.1). Test annotations for both datasets are unavailable and excluded from our experiments. All results are reported on validation sets.

**Models and Baselines.** We evaluate the latest SOTA VLMs: Qwen2.5VL-2B/7B (Bai et al., 2025), LLaVA 1.6-7B (Liu et al., 2024b), InternVL2.5-2B/8B (Wang et al., 2024b) – alongside the task-specific SOTA BEiT3 (Wang et al., 2023). We mainly study the larger models and smaller models performances are reported in Tab. 2. This selection covers both zero-shot VLMs and the task-specific

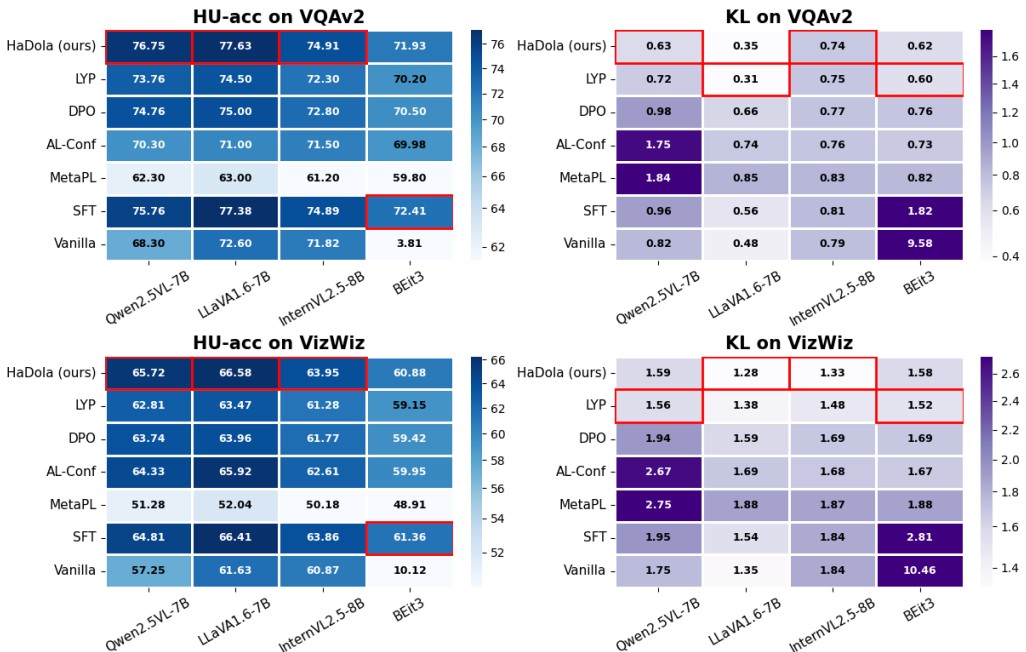

Figure 3: Heatmap comparison of different training methods across four backbone models on VQAv2 and VizWiz, under HU-acc and KL. For HU-acc, darker colors indicate higher accuracy, while for KL, lighter colors indicate smaller divergence. Red boxes highlight the best-performing method for each model. Our method (HaDola) consistently achieves competitive or superior performance across both datasets.

model. For baselines, we consider: (1) VLMs' zero-shot outputs; (2) SFT with LoRA (Hu et al., 2022); (3) Meta Pseudo-labeling (Meta PL) (Pham et al., 2021; Mitra & Soundararajan, 2024), while excluding weaker semi-supervised learning methods like FixMatch (Sohn et al., 2020) and UDA (Xie et al., 2020); (4) Active Learning (AL) for VQA with least-confidence sampling (Karamcheti et al., 2021); (5) DPO training for VQA (Sun et al., 2024); and (6) selective prediction LYP (Dancette et al., 2023). We reasonably do not consider more baselines since to the best of our knowledge, they already span recent VQA training strategies, ensuring fair and extensive comparison. More explanations and baseline setup details are in Appendix C.1.

**Evaluation Metrics.** Traditional VQA evaluation only adopts VQA-accuracy metric. In this work, we assess (i) accuracy via HU-acc (Sec. 4) and (ii) confidence alignment via KL-divergence (Sec. 3). We exclude EntCE (Guo et al., 2017), as prior work (Lan et al., 2025a) shows that it is less effective than KL for VQA.

**Reproducibility and Implementation Details.** We use open-sourced code bases, model checkpoints, and datasets. We provide their information together with all the implementation details, including training in Appendix C.2. Therefore, we emphasize our method and results are highly reproducible.

## 5.2 RESULTS AND DISCUSSION

**Comparison with baselines.** Figure 3 compares HaDola with baselines on the selected metrics and datasets. On accuracy, we observe that for the three zero-shot capable VLMs, HaDola achieves the best performance: with only 5% of labeled data, it even surpasses SFT with 100% annotations. This demonstrates that HU-based supervision outperforms the traditional quantity-based strategy. On BEiT3, which lacks inherent zero-shot ability and heavily relies on large-scale supervision, HaDola does not outperform SFT, but still reaches a comparable level while substantially outperforming all other baselines. On KL divergence, both HaDola and LYP consistently outperform the remaining methods. Between the two, when LYP performs better, HaDola still achieves comparable performance while surpassing all other baselines. This result suggests that leveraging fewer but more informative samples leads to better calibration and thus more reliable models. Compared with LYP, while LYP

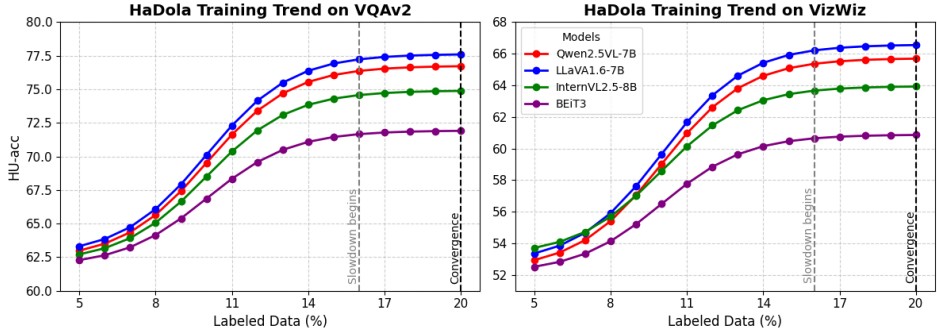

Figure 4: Training dynamics of HaDola. We observe an S-shaped improvement: a rapid increase with 5–10% labels, a slower gain between 10–15%, and convergence after 15%.

| Method (T=15) | VQAv2 (HU-Acc) | | VizWiz (HU-Acc) | |
|---|---|---|---|---|
| | Qwen | LLaVA | Qwen | LLaVA |
| HaDola (full) | **76.75** | **77.63** | **65.72** | **66.58** |
| Replace Selectors with Random Samples | 72.23 | 73.51 | 62.11 | 63.02 |
| Replace Self-Annotate with Human Labels | 73.91 | 75.02 | 64.53 | 64.88 |
| w/o Error Trigger | 71.56 | 72.47 | 60.92 | 61.73 |
| Replace Our Loss with CE | 72.37 | 73.28 | 61.89 | 62.15 |

Table 1: Ablation study on HU-Acc across VQAv2, VizWiz and Qwen2.5VL-7B, LLaVA1.6-7B. We use Qwen and LLaVA for short in the table. Performance consistently drops when replacing or removing each component, showing the necessity of our design.

attains lower divergence by discarding samples it deems hard to answer, HaDola achieves competitive calibration without explicitly removing data, making it more broadly applicable.

**HaDola Training Rounds.** In Figure 4, we present the training dynamics of HaDola on VQAv2 and VizWiz. A clear S-shaped curve emerges across all backbones: models exhibit rapid performance improvement when only 5-10% of the data is labeled, followed by a moderate gain between 10-15%, and eventually plateau after 15%. This finding highlights two important insights. First, substantial gains can be achieved with very limited supervision, showing that HaDola is highly label-efficient. Second, further annotation beyond 15% yields diminishing returns, suggesting that conventional large-scale supervised fine-tuning may lead to unnecessary annotation costs. Together, these results demonstrate that HaDola not only improves absolute performance, but also provides a principled way to reduce the reliance on costly labeled data.

**Ablation Study.** Table 1 presents the ablation results on Qwen2.5VL-7B and LLaVA1.6-7B. We observe consistent performance degradation across both backbones once any component of HaDola is removed or replaced, validating the necessity of our design. Replacing the *Selectors* with random sampling substantially reduces accuracy, confirming that HaDola effectively identifies helpful samples. Substituting *Self-Annotate* with human labels leads to the smallest drop, yet still demonstrates that the model benefits from iteratively self-refining supervision signals beyond human annotations. Removing the *Error Trigger* causes the largest decline, showing that unchecked erroneous labels accumulate and thus harm training. Finally, replacing our tailored loss with standard cross-entropy prevents effective alignment with human preference, yielding suboptimal results. These results highlight that each component plays a complementary and indispensable role in HaDola.

**Effects from different HU degrees.** As shown in Figure 5, when applying simple SFT on the four models, we consistently observe a monotonic trend across both metrics. Models trained on the low-HU subset (L) achieve the best performance, followed by the medium-HU subset (M), while the high-HU subset (H) yields the worst results. This ordering (L > M > H) holds not only on training subsets but also on validation subsets, where lower HU leads to higher accuracy and lower KL divergence. These results suggest that samples with higher HU contribute less to learning, and high-HU is indeed a challenge for VLMs. More results and analysis are in C.3.

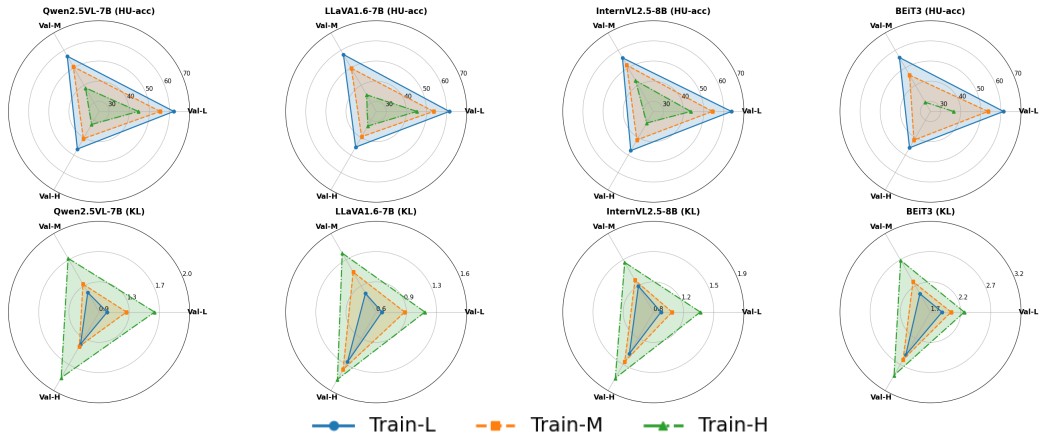

Figure 5: Radar charts of SFT performances across VQAv2 different training and validation subsets with varying HU levels. The upper row shows HU-acc (higher is better), and the lower row shows KL divergence (lower is better) for five models. The three training settings are distinguished by line styles and colors: Train/Val-L, -M, -H means Training or Validation on Low, Medium, or High HU subsets.

## 6 LIMITATION AND CONCLUSION

This work demonstrates that human uncertainty (HU) is not merely a by-product of annotation but a valuable training signal. By systematically analyzing HU, we show that high-HU samples are detrimental, while low- and medium-HU subsets drive both accuracy and calibration. Building on this insight, we propose HaDola, a generalizable framework that integrates HU-aware data across multiple VLMs and two benchmarks. HaDola achieves superior performance with drastically reduced annotation needs. More broadly, our study highlights a new direction for VQA training: rather than indiscriminately scaling labeled datasets, we advocate leveraging human uncertainty as an informative signal to guide both data selection and evaluation, enabling more efficient, better calibrated, and ultimately more human-aligned vision-language models. The main limitation is that HaDola relies on a well-constructed labeled seed set. Although such a seed set is available with limited cost, studying how to use VLMs' zero-shot ability to further reduce the reliance on labeled data will be a worthwhile future direction.

**Ethics statement** We anticipate no ethical concerns with this work. We utilized open-sourced datasets and models, which have been cited.

## 7 ACKNOWLEDGMENTS

Yihong Liu and Hinrich Schütze were supported by the German Research Foundation (DFG, grant SCHU 2246/14-1).

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

## Technical Appendices and Supplementary Material

## A  DIFFERENT HU DEGREES IMPACTS ON VIZWIZ

Similar to Figure 2 (b), we provide comparison of HU-acc and VQA-acc on VizWiz in Figure 6. In general, model performances drops on VizWiz as this is a more challenging dataset, yet we still see a similar trend to VQA's: HU-acc provides a more discriminative view across subsets: models trained on low-, medium-, and high-HU data exhibit a monotonic drop in accuracy as HU grows, indicating that high-HU samples undermine the benefits of SFT on the other two groups. When combining low- and medium-HU data, performance further increases, whereas incorporating high-HU data once again degrades training, confirming their limited utility. In contrast, VQA-acc reflects only small variations and fails to capture the detrimental impact of high-HU samples. This divergence underscores the limitations of frequency-based evaluation and highlights the necessity of HU-aware metrics.

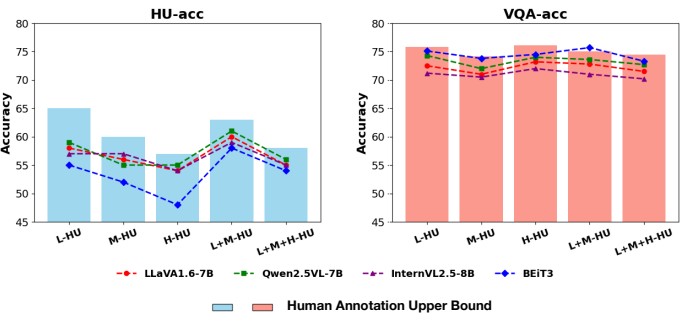

Figure 6: Results on VizWiz: Effects of training samples with different HU degrees, and comparison with VQA-acc. The L, M, H stands for low, medium, and high.

## B  HADOLA SETUP

### B.1  HADOLA PIPELINE FIGURE

Figure 7 provides a clear illustration of HaDola pipeline. Details are introduced in 4.2

### B.2  MORE THEORETICAL DERIVATIONS

**Setup and notation.**  Let $\mathcal{X}$ be the input space and $\Delta^C$ the probability simplex over $C$ answers. For a sample $s \in \mathcal{X}$, let $H(s) \in \Delta^C$ denote the human distribution (HaConf-based), $M_\theta(s) \in \Delta^C$ the current model distribution, and $M_{\mathrm{HU}}(s)$ the HU reference model. Define the supervised risk and calibration gap

$$R(\theta) = \mathbb{E}_{(x,y)}[\mathrm{CE}(y, M_\theta(x))], \qquad \mathcal{C}(\theta) = \mathbb{E}_x\big[D_{\mathrm{KL}}\big(H(x) \,\|\, M_\theta(x)\big)\big].$$

The HaDola objective is

$$\mathcal{L}_{\mathrm{HaDola}}(\theta) = R(\theta) + \beta\,\Phi\big(M_\theta \,\|\, M_{\mathrm{HU}}\big) + \lambda(\mathcal{C}(\theta) - \mathcal{C}(\theta_{\mathrm{HU}}))\,, \qquad (9)$$

where $\Phi\big(M_\theta \,\|\, M_{\mathrm{HU}}\big) = \mathbb{E}_x\big[D_{\mathrm{KL}}\big(M_{\mathrm{HU}}(x) \,\|\, M_\theta(x)\big)\big]$ and $\theta_{\mathrm{HU}}$ are the parameters of $M_{\mathrm{HU}}$.

**Assumptions.**

**A1** (*Smoothness*) For all $x$, the map $\theta \mapsto M_\theta(x)$ is $L$-Lipschitz and $\mu$-smooth in parameters.

**A2** (*Bounded logits*) $\|\log M_\theta(x)\|_\infty \leq B$ uniformly over iterates.

**A3** (*Seed representativeness*) The seed $S_0$ is i.i.d. as $S_r$; its low/medium-HU subsets yield empirical estimates $(\tau_1, \tau_2, h_\omega)$ with sub-Gaussian concentration (Eq. 3 in the main text).

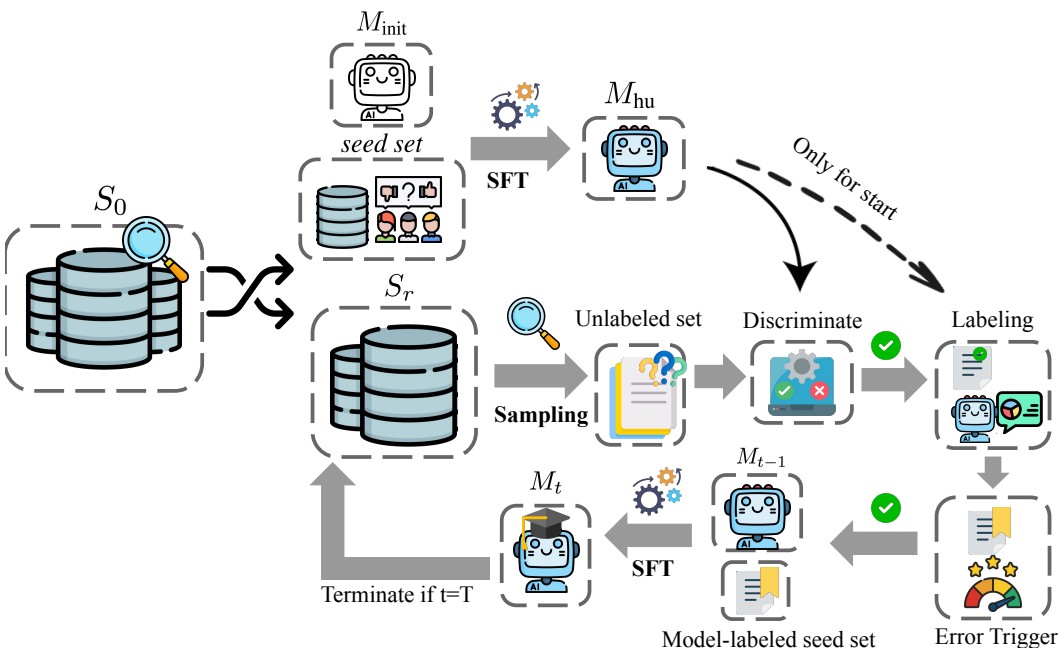

Figure 7: Illustration of HaDola pipelines. Technical details are presented in Section 4.2.

**Stage I (Discriminate): HU window controls distribution shift.** Let $kl_u = D_{\mathrm{KL}}(h_\omega \,\|\, M_t(u))$ for $u \in S_r$. HaDola retains samples with $kl_u \in [\tau_1 - \sigma, \tau_2 + \sigma]$.

**Lemma 1 (Pinsker window).** For any retained $u$,

$$\|M_t(u) - h_\omega\|_1 \le \sqrt{2\,D_{\mathrm{KL}}(h_\omega \,\|\, M_t(u))} \le \sqrt{2\,(\tau_2 + \sigma)}.$$

*Implication.* Discrimination confines training to a total Variation distance-ball around $h_\omega$, excluding high-HU/outlier points. In other words, retained samples are guaranteed to stay within a bounded neighborhood of the human anchor distribution $h_\omega$, so the model only learns from data that is sufficiently consistent with human confidence, while discarding samples that are too uncertain or anomalous to provide reliable supervision.

**Stage II (Self-annotate): pseudo-label bias bound.** Let $\hat{y}_u \sim M_{t-1}(u)$ and $\ell$ be CE. Then

$$\mathbb{E}_{\hat{y}_u}\big[\ell(M_\theta(u), \hat{y}_u)\big] = \mathrm{H}\big(M_{t-1}(u)\big) + D_{\mathrm{KL}}\big(M_{t-1}(u) \,\|\, M_\theta(u)\big).$$

If additionally $\|M_{t-1}(u) - h_\omega\|_1 \le \varepsilon$ (by Lemma 1 at $t-1$), then

$$\left| \mathrm{CE}\big(H(u), M_\theta(u)\big) - \mathrm{CE}\big(M_{t-1}(u), M_\theta(u)\big) \right| \le L_{\mathrm{CE}}\,\varepsilon,$$

for some $L_{\mathrm{CE}}$ depending on $B$ in **A2**. Thus pseudo-label training is a bounded-bias surrogate of HU supervision within the HU window. *Implication.* Self-annotation provides a bounded-bias surrogate of human supervision: as long as samples remain in the HU trust region defined in Stage I, the loss incurred by training on pseudo-labels differs only slightly from that of true HU labels. In other words, even though annotations are generated automatically by the model, they are guaranteed to stay close enough to human confidence distributions to provide reliable training signals.

**Stage III (Error trigger): stability via gradient alignment.** Let $g(u) = \nabla_\theta \ell(M_\theta(u), \hat{y}_u)$ and $g_{\mathrm{ref}} = \mathbb{E}_{(x,y) \in S_0}[\nabla_\theta \ell(M_\theta(x), y)]$. If $\cos(g(u), g_{\mathrm{ref}}) \ge \tau_g > 0$ and a TracIn-mini score $s_{\mathrm{tracin}}(u) \le \tau_t < 0$, the contribution to validation loss is non-increasing (first-order view).

Under **A1** and step size $\eta \le 1/L$, updating on samples passing both triggers yields

$$\Delta L_{\mathrm{val}} \le -\eta\,\kappa + O(\eta^2),$$

for some $\kappa > 0$ proportional to $\mathbb{E}[\langle g(u), \nabla L_{\text{val}} \rangle]$ over retained samples.

*Implication.* The error trigger ensures stability during iterative self-training: pseudo-labeled samples are only retained if their gradient directions are consistent with human-labeled data and their global influence does not degrade validation performance. In other words, this mechanism prevents harmful pseudo-labels from accumulating, so that each added sample contributes constructively to model refinement rather than introducing noise.

**Stage IV (Training): decomposition and effect.** From equation 9, $\mathcal{C}(\theta_{\text{HU}})$ is constant; minimizing $\mathcal{L}_{\text{HaDola}}$ equals minimizing

$$R(\theta) + \beta\,\Phi(M_\theta \,\|\, M_{\text{HU}}) + \lambda\,\mathcal{C}(\theta).$$

The last term penalizes misalignment to $H$, while the middle term constrains drift from the HU reference.

**Lemma 2 (Relative-KL improvement).** At any stationary point $\hat{\theta}$ with $\beta, \lambda > 0$, either $M_{\hat{\theta}} = M_{\text{HU}}$ a.e., or $\mathcal{C}(\hat{\theta}) < \mathcal{C}(\theta_{\text{HU}})$ on a set of non-zero measure.

*Implication.* The HU-aware loss jointly balances accuracy, stability, and human alignment: cross-entropy drives correct predictions, HU regularization keeps the model anchored to the reference distribution, and the relative KL term explicitly pushes the model to better approximate human uncertainty. In other words, this objective prevents the model from drifting while encouraging calibration beyond the HU reference, yielding VLMs that are not only accurate but also faithfully reflect human confidence distributions.

*Proof sketch.* Take the inner product of the first-order condition $\nabla R + \beta \nabla \Phi + \lambda \nabla \mathcal{C} = 0$ with $\log(H/M_\theta)$ and use the Bregman structure of $D_{\text{KL}}$.

**Putting it together.** Combining Lemma 1 (HU-window), the bounded-bias surrogate view, Proposition 1 (stability), and Lemma 2 (relative-KL improvement) gives:

Under **A1–A3**, fixed window $[\tau_1 - \sigma, \tau_2 + \sigma]$, step size $\eta \leq 1/L$, and $\beta, \lambda > 0$, the HaDola iterates $\{\theta_t\}_{t=0}^T$ satisfy

$$\mathcal{L}_{\text{HaDola}}(\theta_{t+1}) \leq \mathcal{L}_{\text{HaDola}}(\theta_t) - \eta\,\kappa_t + O(\eta^2), \qquad \mathbb{E}\big[\mathcal{C}(\theta_T)\big] < \mathcal{C}(\theta_{\text{HU}}),$$

and $R(\bar{\theta}_T) \leq \min_{t \leq T} R(\theta_t) + O(1/T)$ with $\bar{\theta}_T = \frac{1}{T}\sum_{t=1}^T \theta_t$. Thus HaDola decreases prediction error and improves human alignment beyond the HU reference while maintaining stability.

### B.3   PARAMETERS AND THRESHOLDS

**Threshold estimation.** Both thresholds $\tau_g$ and $\tau_t$ are derived from the low- and medium-HU subsets of the seed set $S_0$. Formally, let $s_g(x, y; \theta_t)$ denote the gradient consistency score of sample $(x, y)$ and $s_{\text{tracin}}(x, y; \theta_t)$ its TracIn-mini score. We define

$$\tau_g = \mathbb{E}_{(x,y) \in l_{S_0} \cup m_{S_0}}\big[s_g(x, y; \theta_t)\big], \qquad \tau_t = \mathbb{E}_{(x,y) \in l_{S_0} \cup m_{S_0}}\big[s_{\text{tracin}}(x, y; \theta_t)\big]. \tag{10}$$

A pseudo-labeled sample $(u, \hat{y}_u)$ is retained only if

$$s_g(u, \hat{y}_u; \theta_t) \geq \tau_g \quad \wedge \quad s_{\text{tracin}}(u, \hat{y}_u; \theta_t) \leq \tau_t. \tag{11}$$

In other words, $\tau_g$ ensures sufficient gradient alignment with human-labeled data, while $\tau_t$ enforces that the pseudo-labeled sample does not harm validation performance.

**(iii) Default values.** Robust defaults are $\beta \in [0.1, 1]$ (e.g., 0.3) and $\lambda \in [0.1, 2]$ (e.g., 0.5–1.0). We monitor HU-acc and KL during training: if accuracy rises but KL stagnates, we increase $\lambda$; if KL improves but accuracy drops, we reduce $\lambda$ or strengthen $\beta$.

### B.4   COMPUTATIONAL EFFICIENCY.

HaDola introduces no additional training overhead compared to standard SFT. All components (discrimination, self-annotation, error trigger, and loss) reuse existing forward and backward gradient.Unlike standard SFT, which repeatedly trains on 100% of annotated data each epoch, HaDola

| Model | OKVQA | | GQA | |
|---|---|---|---|---|
| | Vanilla | HaDola (trained on VQAv2) | Vanilla | HaDola (trained on VQAv2) |
| Qwen2.5VL-2B | 62.6 | **64.5** | 72.1 | **79.8** |
| LLaVA1.6-7B | 61.2 | **63.7** | 68.4 | **75.1** |
| InterVL2.5-2B | 59.8 | **61.0** | 66.5 | **73.2** |
| BEiT3 | 2.4 | **59.1** | 59.1 | **61.5** |

Table 2: Performance comparison of HaDola and baselines on OKVQA and GQA datasets on HU-Acc.

only uses 5–20% of the data per round (starting from a 5% seed and incrementally adding 1% per iteration). This drastically reduces the effective training load while achieving superior accuracy and calibration, so our framework incurs *lower computational cost* than SFT. A detailed quantification of computational cost is beyond the scope of this work, since our focus is on the effect of human uncertainty on training and calibration, rather than runtime profiling. Thus, our method achieves better accuracy and calibration without extra computational cost.

## C  MORE EXPERIMENTAL DETAILS AND RESULTS

### C.1  DATASETS AND BASELINES SETUP

**Datasets.**  We select VQAv2 and VizWiz as our main datasets because they are, to the best of our knowledge, the only open-source VQA datasets that explicitly provide human confidence annotations, which are indispensable for studying HU. Other widely used datasets such as OKVQA (Marino et al., 2019) or GQA (Hudson & Manning, 2019) do not include HU information, and thus cannot directly support our investigation. Nevertheless, to further test the generality of HaDola, we also evaluate models trained on VQAv2 in a cross-domain setting, and observe that HaDola retains strong performance when transferred to other VQA-style datasets, demonstrating its domain generalization ability. Table 2 demonstrates that HaDola consistently surpasses all baselines across both datasets and evaluation metrics. In particular, it suggests that the benefits of HaDola are not restricted to a single benchmark but generalize across distinct domains. This generalization ability highlights that our design choices—especially the integration of HU-aware training signals—enable the model to transfer effectively across datasets with different challenges, thereby ensuring robustness and broader applicability.

**Baselines.**  We compare HaDola with a broad set of representative training strategies, including *supervised learning*, *semi-supervised learning*, *active learning*, *reinforcement learning-based data augmentation*, and *selective prediction*. To the best of our knowledge, these baselines cover all major VQA training paradigms that allow fair comparison with our setting. We therefore reasonably do not consider additional models, as our selection already provides a sufficiently comprehensive and fair evaluation.

### C.2  REPRODUCIBILITY AND IMPLEMENTATION DETAILS.

Our work is highly reproducible. All code, pre-processing scripts, and detailed hyperparameters will be released after the anonymous period.

Details include: (i) model architectures and checkpoints are from the following open-sourced code bases: Qwen2.5VL [1], LLaVA 1.6 [2], InternVL2.5 [3], and BEiT3 [4]. (ii) dataset splits are from the official website: VQAv2 [5], VizWiz [6]. (iii) training hyperparameters (learning rates, batch sizes, optimizers, number of rounds $T$, etc, in Tab. 3:

---

[1] https://github.com/QwenLM/Qwen2.5-VL, https://github.com/sandy1990418/Finetune-Qwen2.5-VL

[2] https://github.com/haotian-liu/LLaVA, https://github.com/arielnlee/LLaVA-1.6-ft

[3] https://huggingface.co/OpenGVLab/InternVL

[4] https://github.com/microsoft/unilm/tree/master/beit3

[5] https://visualqa.org/

[6] https://vizwiz.org/tasks-and-datasets/vqa/

| Parameter | Value |
|---|---|
| Hardware | $8 \times$ A100 80G (single node) |
| Precision | BF16 |
| Optimizer | AdamW (weight decay: 0.01 or 0) |
| Learning rate (LLM) | $1 \times 10^{-6}$ |
| Learning rate (Vision / Projector) | $1 \times 10^{-6}$ / $1 \times 10^{-5}$ |
| Batch size (effective) | 48–64 |
| Gradient accumulation | 4 (when batch per device $<$ global batch) |
| Epochs | 3 for VLMs, 10 for BEiT3 |
| Warmup ratio | 0.03–0.1 |
| Scheduler | Cosine |
| Max text length | 2048–4096 |
| Gradient checkpointing | Enabled |
| LoRA | Rank 16, $\alpha$=32 |

Table 3: Core training configurations used in our experiments.

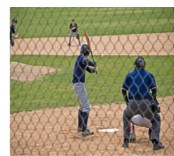 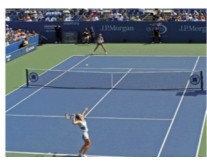 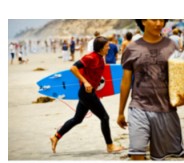 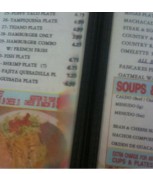

(a)
Q: how many people are there?
Human: 5
SFT: 3
HaDola: 5
HaDola w/o our loss: 4

(b)
Q: who is on offense?
Human: person at bottom
AL: woman
HaDola: bottom person
HaDola w/o self-annotate: man

(c)
Q: what color is the surfboard?
Human: red and blue
MetaPL: blue
HaDola: red and blue
HaDola w/o error trigger: blue

(d)
Q: how much is the shrimp plate?
Human: 6.75
Vanilla: 6
HaDola: 6.75
HaDola w/o selector: 615

Figure 8: HaDola's generation (also as self-label) together with case study.

### C.3 MORE EXPERIMENTAL RESULTS AND ANALYSIS

**Self-annotated label Analysis & Case Study.** To disentangle the effect of each design choice, we conduct an ablation study on HaDola. HaDola's outputs serve as both answers (final round) and self-annotated labels for middle rounds. As shown in Figure 8, compared with baselines, HaDola consistently yields correct and human-aligned results. Removing any component leads to clear degradations: the model drifts away from human preferences while the self-annotated iterative supervision weakens, where obvious mistakes are left uncorrected. These results confirm that each design choice is indispensable for HaDola's superiority.

**Effects from different HU degrees.** Consistent with our earlier findings on VQAv2, we again observe under simple SFT a clear monotonic trend across both accuracy and KL divergence on all four models. Training on low-HU subsets (L) yields the strongest performance, medium-HU subsets (M) perform worse, and high-HU subsets (H) degrade performance the most. This ordering (L > M > H) appears consistently on both training and validation subsets, indicating that higher HU samples provide less useful supervision and that high-HU remains a fundamental challenge for VLMs.

**Size selection of seed set.**

To further analyze the effect of seed set size, we vary it from 1% to 10% and report Hu-acc results in Table 4. As shown in the table, all models on both datasets exhibit rapid gains when the seed set increases from 1% to 3%, followed by slower improvements and eventual convergence after round 5. This indicates that the majority of performance benefits are already captured in the early stages, while adding more labeled data beyond 5% yields only marginal gains. Therefore, in the remainder of this work we choose 5% as the default seed set size, as it strikes a favorable balance between model performance and annotation cost. Moreover, a 5% budget is also more realistic and practical in real-world applications Schlegel et al. (2025), where annotation budgets are constrained in both industry and research settings, and stands also for text-based tasks Chen et al. (2025); Xue et al. (2023); Lan et al. (2023; 2022).

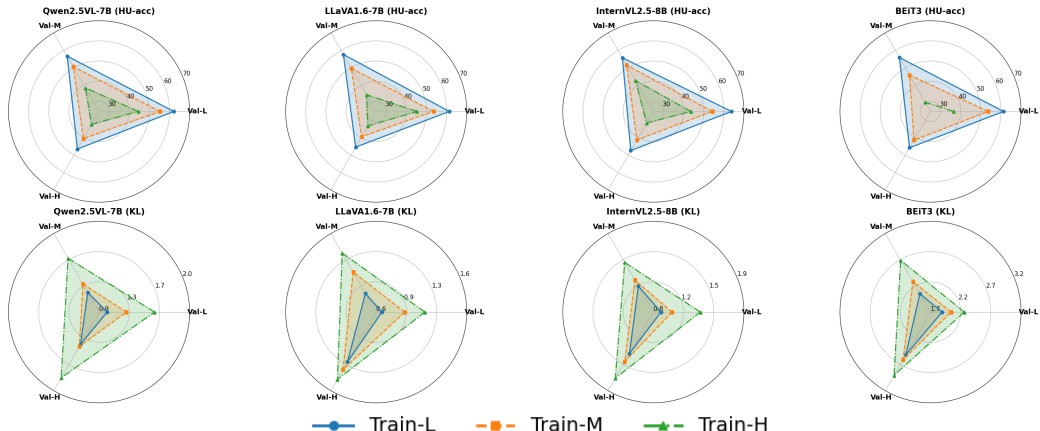

Figure 9: Radar charts of SFT performances across VizWiz different training and validation subsets with varying HU levels. The upper row shows HU-acc (higher is better) and the lower row shows KL divergence (lower is better) for five models. The three training settings are distinguished by line styles and colors: Train/Val-L, -M, -H means Training or Validation on Low, Medium, or High HU subsets.

| Base Model | Dataset | 1% | 2% | 3% | 4% | 5% | 6% | 7% | 8% | 9% | 10% |
|---|---|---|---|---|---|---|---|---|---|---|---|
| Qwen2.5VL-7B | VQAv2 | 0.62 | 0.68 | 0.71 | 0.73 | 0.77 | 0.77 | 0.77 | 0.77 | 0.78 | 0.78 |
| | VizWiz | 0.55 | 0.60 | 0.63 | 0.65 | 0.66 | 0.66 | 0.66 | 0.66 | 0.67 | 0.67 |
| LLaVA1.6-7B | VQAv2 | 0.60 | 0.66 | 0.70 | 0.73 | 0.75 | 0.76 | 0.76 | 0.76 | 0.77 | 0.77 |
| | VizWiz | 0.54 | 0.59 | 0.62 | 0.64 | 0.67 | 0.67 | 0.67 | 0.67 | 0.67 | 0.68 |
| InternVL2.5-8B | VQAv2 | 0.58 | 0.64 | 0.68 | 0.71 | 0.72 | 0.73 | 0.73 | 0.73 | 0.74 | 0.74 |
| | VizWiz | 0.52 | 0.57 | 0.61 | 0.63 | 0.64 | 0.64 | 0.64 | 0.64 | 0.65 | 0.65 |
| BEiT3 | VQAv2 | 0.45 | 0.58 | 0.64 | 0.68 | 0.72 | 0.72 | 0.72 | 0.73 | 0.73 | 0.73 |
| | VizWiz | 0.39 | 0.45 | 0.53 | 0.60 | 0.61 | 0.61 | 0.61 | 0.61 | 0.62 | 0.62 |

Table 4: HaDola top performance (HU-acc) across seed set sizes (1–10%) on VQAv2 and VizWiz. We observe steep gains during the first three rounds, rapid improvement up to round 5, and convergence afterwards.

**Comparison with traditional Calibration**

Finally, we briefly compare HaDola with traditional calibration methods, in particular Temperature Scaling (TS). We emphasize that this work is not a calibration paper and HaDola is not designed solely for calibration; rather, our focus is on leveraging HU for data selection and training. Nevertheless, it is informative to analyze HaDola's improvements in calibration relative to TS. We also note that several recent works have studied calibration (Wieczorek et al., 2025; Eisenschlos et al., 2024; Lan et al., 2025a). However, these studies primarily address reliability and human distribution alignment without evaluating model accuracy, and thus do not provide a fair comparison to our setting. Among them, Lan et al. (2025a) specifically evaluates TS for VQA, while the other two works have not yet released code, preventing a direct implementation for comparison.

As shown in Table 5, HaDola consistently achieves the lowest KL divergence across all four base models and both datasets, demonstrating clear advantages in calibration. In contrast, Temperature Scaling (TS, $T$=1.2) only partially reduces KL: while it slightly improves calibration on VQAv2 (e.g., Qwen2.5VL-7B and LLaVA1.6-7B), it even degrades performance on VizWiz, where KL values increase compared to the vanilla baseline. This highlights the instability of post-hoc calibration under domain shift or noisy annotations. In all cases, HaDola substantially outperforms TS, showing stable and large improvements. The effect is particularly striking on BEiT3, where the vanilla model exhibits extremely poor calibration (KL over 9.5 and 10.4), TS reduces the gap but still remains high, while HaDola lowers KL to 0.62 and 1.58, reaching the same level as recent VLMs. These results confirm that HaDola not only surpasses post-hoc calibration but also achieves robust and reliable

| Base Model | VQAv2 | | | VizWiz | | |
|---|---|---|---|---|---|---|
| | Vanilla | TS | HaDola | Vanilla | TS | HaDola |
| Qwen2.5VL-7B | 0.82 | 0.67 | **0.64** | 1.75 | 2.45 | **1.59** |
| LLaVA1.6-7B | 0.48 | 0.41 | **0.35** | 1.35 | 2.52 | **1.28** |
| InternVL2.5-8B | 0.79 | 0.77 | **0.74** | 1.84 | 2.58 | **1.33** |
| BEiT3 | 9.58 | 2.221 | **0.62** | 10.46 | 2.80 | **1.58** |

Table 5: KL divergence (lower is better) for Vanilla, Temperature Scaling (TS, $T$=1.2), and HaDola across two datasets.

calibration improvements across diverse models and datasets. This suggests that explicitly leveraging HU during training is more effective than applying post-hoc calibration methods such as TS.

