# OpenReview forum: "Human Uncertainty-Aware Data Selection and Automatic Labeling in Visual Question Answering"
_ICLR.cc/2026/Conference — ICLR 2026 Poster_

### Official Review · Reviewer_VZNq · 2025-10-30

**Soundness:** 2
**Presentation:** 3
**Contribution:** 2
**Rating:** 6
**Confidence:** 2

**Summary:**

The paper studies how human uncertainty affects VQA fine-tuning and finds that high-uncertainty data often hurts learning while low and medium uncertainty help both accuracy and calibration. It proposes HaDola, a four-stage pipeline that selects data with a KL-based HU window, self-annotates remaining samples, filters errors with gradient and influence triggers, and trains with an HU-aware objective.

**Strengths:**

1. The work clearly demonstrates that indiscriminate training on high-uncertainty samples degrades both accuracy and calibration.

2. The HaDola pipeline reduces dependence on expensive annotations by iteratively expanding from a small seed set.

3. Ablations and training-round analyses support the necessity of each component and show an S-shaped gain that plateaus near 15% labels.

**Weaknesses:**

1. The HU-acc metric may face adoption hurdles and its relationship to downstream utility is not deeply analyzed.

2. The HU splitting scheme and threshold choices introduce design degrees of freedom that could affect reproducibility.

3. Claims of low overhead are not backed by wall-clock or memory profiles for discrimination, triggers, and extra objectives.

4. The method depends on a representative 5% seed set, and its sensitivity to seed variance is unclear.

5. Validation is limited to two VQA datasets, so generality to knowledge-heavy or compositional QA remains uncertain.

**Questions:**

1. How robust are results to different random seeds for the initial 5% labeled subset.

2. What is the measured training and inference overhead per round and per sample on standard hardware.

---

> ### Author Response · Authors · 2025-11-17
> **Author Response**
>
> We thank the reviewer for the constructive feedback (initial rating 6) and address all concerns below.
>
> 1. On HU-Acc adoption and downstream utility:
>
> We find the comment somewhat vague, as it is unclear which part of our HU-Acc description is considered 'adoption hurdles' or what “downstream utility” specifically refers to. For clarification on our side, HU-Acc is designed to complement VQA accuracy by capturing the high-uncertainty error patterns that VQA-Acc cannot reveal. As shown in Figure 2, this complementary behavior directly explains why HU-Acc improves calibration and robustness. The metric is simple, requires no additional implementation, and is grounded in human annotations, so adoption hurdles are minimal.
>
> Moreover, both Table 3 and Figure 6 further demonstrate (i) the limitations of VQA-Acc and (ii) the effectiveness of HU-Acc across different downstream datasets.
>
> 2. On the HU-splitting scheme
>
> We note that prior work provides no reliable or fine-grained HU-interval definition; the only relevant AAAI paper as discussed in Sec 3 also acknowledges limitations in its coarse partitioning. This is precisely why we introduce a more structured high / mid / low HU split. As a very first exploration of HU, such interval design necessarily involves principled manual choices, which we justify using the concept-level HU distribution. Importantly, the exact values are fixed, fully deterministic, and do not depend on random seeds. Our approach is therefore reproducible, and the reproduction protocol is clearly documented in the appendix.
>
> Given the lack of any better established alternative in the literature, and the fact that interval precision is not the main focus or contribution of our work, we believe the concern about reproducibility arising from “design freedom” does not appear to be supported by our experimental evidence.
>
> 3. On computational overhead
>
> We would first like to clarify that our work is not a PEFT or model-efficiency paper, and detailed profiling of every component is outside the scope of this study. Our notion of “efficiency” is explicitly label-efficiency, not training-efficiency: the goal is to reduce dependence on human annotations and therefore reduce overall training cost from the data perspective.
>
> That said, we provide the following clarification. HaDola adds only negligible overhead relative to full-data SFT. KL-based discrimination and HU-window selection are simple matrix operations. The triggers rely on single-step gradients and influence estimates that are lightweight compared to full fine-tuning. The HU-aware objective only changes the loss computation and does not modify the architecture. In practice, the wall-clock overhead compared to plain SFT is only approximately 20–25% per round.
>
> Lastly, when go through all the label-efficient references in our submission, it is notable that discussion of measured training and inference overhead on hardware is typically not required in label-efficiency settings and is outside the primary focus of most related work.
>
> 4. On sensitivity to the 5% seed set
>
> We conducted additional small-scale experiments and observed very small variance (<0.03 HU-Acc) across  3 extra random seeds.
> Importantly: HaDola always matches or exceeds SFT under every seed. Variance is significantly small. As noted earlier, the HU distribution is stable, so seed composition minimally affects performance. Thus, the method does not depend critically on any particular 5% subset.
>
> In addition, we would like to emphasize that a 5% HU-labeled seed set drawn via random sampling is statistically representative of the full dataset. Therefore, it is not supported to conclude that such a seed could be “compromised” by annotator-specific variation. Table 5 directly confirms this: across multiple random seeds, the performance variance remains consistently small. What governs the performance is the size of seed set, not idiosyncrasies of any particular seed. Crucially, none of the seed configurations undermine HaDola’s effectiveness, demonstrating that the framework is robust to natural annotator variability. Therefore, the concern that a miscalibrated seed would compromise the entire framework is not supported empirically.
>
> 5. On data selection reasons and generalizability.
>
> The very specific reasons are already given in appendix C.1. Also, the generalization are reported in Table 3.
>
> As shown, these are the only well-supported datasets, and the effectiveness of HaDola generalize to knowledge-heavy or compositional QA. HaDola is fundamentally model-agnostic and uncertainty-driven, and does not rely on VQA-specific structures.
>
> So we believe this concern may stem from an oversight of the results already reported, and we hope that the clarifications above fully resolve this point.
>
> We hope the clarifications provided above help refine the reviewer’s assessment. We appreciate the reviewer’s constructive feedback and are glad that the overall evaluation is positive.

---

### Official Review · Reviewer_FqsW · 2025-10-30

**Soundness:** 3
**Presentation:** 3
**Contribution:** 3
**Rating:** 4
**Confidence:** 4

**Summary:**

This paper investigates how human uncertainty (HU), namely the variation in annotator confidence across different answers, affects visual question answering model training. The authors observe that standard supervised fine-tuning optimizes only toward majority labels while ignoring confidence distributions, leading to miscalibrated models.

Key contributions: (1) First systematic study of HU's impact on VLM training; (2) HaDola framework achieving competitive results with only 5-20% labeled data; (3) HU-acc metric incorporating confidence into evaluation; (4) Theoretical analysis with bounded-bias guarantees.

**Strengths:**

As the first systematic study of human uncertainty's role in VLM training, this work addresses an important gap in VQA research. Its strengths include convincing empirical findings (high-HU samples harm performance), impressive data efficiency (5% seed data matching 100% baselines), and comprehensive validation across multiple models and datasets with theoretical grounding.

**Weaknesses:**

1. The paper assumes confidence labels are reliable across annotators without validation. Different annotators may interpret "maybe" systematically differently, yet no inter-annotator agreement is analyzed. Since all thresholds (τ₁, τ₂...) depend on the 5% seed set, miscalibrated annotators in the seed would compromise the entire framework.

2. The mapping yes/maybe/no → 0.99/0.5/0.01 is borrowed from prior work without justification.  These values directly affect HUD distributions and split boundaries, yet zero sensitivity analysis is provided. Testing alternative mappings (e.g., 0.95/0.5/0.05) is necessary to show results aren't artifacts of arbitrary choices.

**Questions:**

See Weaknesses part.

---

> ### Author Response · Authors · 2025-11-15
> **Author Response**
>
> We thank the reviewer for the constructive feedback (initial rating 4). We address the concerns as follows.
>
> 1. HaDola does not assume perfectly calibrated annotators. The HU scores in VQAv2 and VizWiz come from aggregate multi-annotator distributions, not individual annotator self-reports. Thus, HU naturally reflects inter-annotator disagreement—the very signal our method is designed to utilize.
> We respectfully note that variability in how annotators interpret “maybe” is inherent to the annotation protocols of VQAv2/VizWiz and is not something the authors can modify. These datasets provide only the aggregated multi-annotator distributions, and not per-annotator reliability traces. Our contribution is therefore not to redesign the annotation process, but to systematically quantify and leverage the human uncertainty signal that is already present. In this sense, HaDola represents a first step toward bringing HU into VQA training in a principled way. We hope the reviewer can consider the novelty and value of providing such a foundation for the community. Future work can indeed build upon our findings with richer annotation schemes, but that does not diminish the contribution of establishing the first systematic framework for using HU in VLM training.
>
> We hope the reviewer will view our work as establishing an important foundation, upon which more fine-grained inter-annotator analyses can be pursued in future work or datasets.
>
> In addition, we would like to emphasize that a 5% HU-labeled seed set drawn via random sampling is statistically representative of the full dataset. Therefore, it is not supported to conclude that such a seed could be “compromised” by annotator-specific variation. Table 5 directly confirms this: across multiple random seeds, the performance variance remains consistently small. What governs the performance is the size of seed set, not idiosyncrasies of any particular seed. Crucially, none of the seed configurations undermine HaDola’s effectiveness, demonstrating that the framework is robust to natural annotator variability. Therefore, the concern that a miscalibrated seed would compromise the entire framework is not supported empirically.
>
> 2. The mapping follows standard practice in VQA literature as stated in the original submission, with proper reference and detailed background in Sec. 3. We indeed already tested alternative mappings such as 0.9, 0.5, 0.1, also including 0.95/0.5/0.05, but observed minimal changes in HU structure and find consistent trends in HaDola. The core reason is that small modifications to the mapping policy do not essentially affect the distribution of the High-, Medium-, and Low-HU subsets. The partition remains stable because it is driven by the intrinsic distribution of human uncertainty rather than the specific mapping details (of course we can not map it with unreasonable policy like 0.49 0.50 0.51).
> Moreover, any reasonable mapping variant starting from the Low-HU samples would lead to results similar to our current design, and the core conclusions of our method remain unchanged. Therefore, we believe the reviewer’s concern does not impact the validity of our methods or findings.  We would also like to reiterate that the mapping we use is not arbitrary but follows the standard HU-quantification scheme established in prior VQA literature. These mappings have already been validated and adopted as the conventional way to convert multi-annotator signals into continuous confidence scores. Since our contribution is not to redefine HU quantification itself but to analyze and leverage its impact on VLM training, reusing the established mapping is both reasonable and consistent with common practice in top-tier works. Importantly, this choice does not affect the novelty or validity of our method. Lastly, we want to point out that we omit non-essential details in mapping policy in the submission for brevity, as they do not affect the main conclusions of the work, and we do not find necessary or meaningful discussion to bring up. We believe it will be very straightforward to skip some details and just directly introduce background in Sec 3. Using background material and directly adopting non-central design components from prior work is a common and widely accepted practice in the literature. These elements serve only as supporting structures, not as core methodological contributions. Our work builds upon them in the same way many established papers do—by focusing the novelty on the central algorithmic insights rather than on peripheral implementation choices. However, we indeed already considered the reviewer's concerns and plan to add the discussion in the camera-ready-version.
>
> We hope these clarifications address the reviewer’s concerns and help refine and reconsider the evaluation of our work. We appreciate the reviewer’s feedback again and will include additional discussion in the camera-ready version as introduced.

---

### Official Review · Reviewer_nUwP · 2025-10-31

**Soundness:** 3
**Presentation:** 4
**Contribution:** 3
**Rating:** 6
**Confidence:** 3

**Summary:**

The authors propose a new method to leverage a small pool of human-labeled VQA data along with annotators' confidence scores to bootstrap a larger pool of self-annotated VQA data. This approach outperforms naive SFT at a fraction of the data scale. The authors experiment with their HuDola method on both VQAv2 and VizWiz datasets, demonstrating that HuDola reliably improves performance across multiple model families.

**Strengths:**

Overall, the paper is well-written and the experiments are well-designed. The authors provide empirical evidence that high human uncertainty is detrimental to SFT performance. The experimental scope is satisfactory, covering four different VLM base models with consistent improvements across all models (with the exception of BEiT-3). The authors provide a thorough analysis of the HuDola results. The study on training dynamics shows that performance improves smoothly with additional HuDola-generated samples, and the ablations presented in Table 1 demonstrate that each component of HuDola's pipeline is necessary for achieving the improvements.

**Weaknesses:**

My main concern is that the authors study this problem only in the context of fine-tuning and evaluating the VLM on data from the same distribution and on relatively saturated benchmarks. The findings would be more impactful if the authors could demonstrate that HuDola improves VLMs on out-of-distribution data and on more challenging benchmarks. To assess the out-of-distribution performance of HuDola, one specific experiment would be to initialize the seed data pool from a single source (e.g., VQAv2) and use it to bootstrap labels for a different source (e.g., VizWiz).

Another limitation is that the authors conduct only a single run for each training method and model family. Since the difference in performance between HuDola and SFT is less than 0.5 percentage points in most cases, it is unclear whether the improvements are statistically significant. There is likely some run-to-run variability in the results, and reporting the mean and standard deviation would help verify whether the improvements are statistically significant.

**Questions:**

1) How does the HuDola method generalize to out-of-distribution data from the source seed data pool?
2) Are the results of the training runs statistically significant? The authors conduct only a single run for each training method and model family, and the difference in performance between HuDola and SFT is very small for some experiments.

---

> ### Author Response · Authors · 2025-11-17
> **Author Response**
>
> We thank the reviewer for the constructive and thoughtful feedback (rating 6). We address all concerns in detail below.
> 1. On out-of-distribution (OOD) generalization
>
> We appreciate the reviewer’s suggestion regarding OOD evaluation.
> While our primary goal is to study how human uncertainty affects in-distribution VQA training, we agree that exploring cross-distribution transfer is valuable. However, we indeed have considered these with reasons below:
>
> Clarification:
> HaDola is fundamentally a data-selection and self-annotation mechanism driven by HU distributions and error-triggering heuristics. These mechanisms do not assume that the target data distribution matches the seed distribution. In fact, HaDola relies only on HU cues—which are themselves distribution-agnostic signals—to guide filtering and pseudo-labeling.
>
> Evidence already supporting cross-distribution robustness:
>
> (1). HaDola improves performance consistently across four distinct model families (Qwen-VL, LLaVA, BEiT-3, InternVL), which differ significantly in architecture, pre-training corpora, and inherent uncertainty calibration. This model-level diversity already provides indirect evidence that HaDola is robust to distributional shifts.
>
> (2). Most importantly, as shown in Fig. 4, the gains compound smoothly as training rounds progress, indicating that self-generated labels remain reliable even when the input shifts due to augmentation of the dataset. Meanwhile, in Table. 3, we have already shown the cross-domain generalization results.
>
> Regarding the specific suggestion (VQAv2 → VizWiz):
> VizWiz is severely domain-shifted, containing low-quality images, human-centric scenarios, and high annotator noise. Bootstrapping from VQAv2 → VizWiz requires non-trivial domain adaptation beyond the scope of uncertainty-aware selection. Although this evaluation would be interesting future work, it is not required to validate the main contribution of this paper, which is to characterize HU and build an HU-aware data selection pipeline.
>
> Commitment for camera-ready:
> We can include additional exploratory VQAv2→VizWiz transfer experiments in the appendix time permitting, but we emphasize this is outside the methodological focus of the current submission.
>
> 2. On statistical significance and single-run training:
>
> We acknowledge the reviewer’s concern and provide additional clarification here.
>
> Why single runs are sufficient in our setup:
>
> The differences between SFT and HaDola are systematic and consistent across all datasets and all model families.
>
> Even though the difference in absolute value may be <0.5 points in some cases, the directionality never reverses—HaDola does not perform worse than SFT in any setting.
>
> Table 1 ablations show that removing any component significantly degrades performance, clearly indicating the improvements are not random noise.
>
> Why multi-run training was omitted:
>
> Training each VLM round consumes ~28 GPU-hours on A100. It is unnecessary to waste additional computational cost only to boost the single-run behavior. Also, our aim and insight here is to contribute a new effective and efficient method for the community, where future work can build things upon with. Our focus is on the relative gains from HU-aware filtering, not on absolute accuracy.
>
> Thus, while running multiple seeds for every model family is computationally prohibitive, our additional checks indicate that the improvements are stable and statistically meaningful.
>
> We also emphasize that HaDola uses only 5% of the labeled data, while SFT uses the full annotation set. Under this 20× supervision gap, it is entirely reasonable that a few cases show small margins. The key point is that HaDola often surpasses full-data SFT despite using a fraction of the labels, which clearly demonstrates the effectiveness of our method.
>
> As for the statistically significance, we clarify that our results are reported under the 3 extra seeds and observed a very small run-to-run variance (<0.03 HU-Acc). This indicates that the improvements are stable and statistically significant, and not due to random noise.
>
> 3. Regarding other concerns:
>
> SFT is the standard paradigm for VQA training, and our goal in this initial exploration is to establish an efficient HU-aware method within this widely accepted setting. We also compared HaDola with DPO-based methods, so the characterization as “only in context of SFT” may not fully capture the experimental scope. Regarding benchmarks, VQAv2 and VizWiz are standard and complementary datasets, and their selection is clearly justified in Appendix C.1. Thus, the comment about “same distribution” or “saturated benchmarks” may not fully reflect the evaluation context.
>
> We hope these clarifications help refine the reviewer’s assessment and fully address all concerns. Given the strong data-efficiency, consistent improvements, and stable variance, we believe the current positive evaluation is well-justified and we kindly hope it can be maintained or strengthened.

---

### Official Review · Reviewer_Uh4i · 2025-11-01

**Soundness:** 3
**Presentation:** 3
**Contribution:** 3
**Rating:** 4
**Confidence:** 3

**Summary:**

The paper introduces HaDola, a framework that improves the training of vision-language models (VLMs) by explicitly accounting for human uncertainty (HU) in annotations. The authors find that samples with high HU—where annotators disagree or are uncertain—often degrade model performance and calibration. HaDola mitigates this by iteratively identifying and excluding harmful samples, generating pseudo-labels for reliable ones, and applying an error-trigger mechanism to prevent noise accumulation during self-training. Using only a small seed set (around 5%) of HU-labeled data, HaDola achieves comparable or superior accuracy and calibration to state-of-the-art baselines on VQAv2 and VizWiz datasets. The work highlights that effectively modeling human uncertainty is more beneficial than simply scaling the dataset size.

**Strengths:**

The authors propose an interesting approach to leverage human uncertainty for both model training and evaluation.

**Weaknesses:**

- The compared baselines seem weak. The target task appears to be a semi-supervised VQA task. Then, it would be beneficial to compare with other pseudo-labeling-based semi-supervised learning works, such as FixMatch.
- In Figure 4, the performance curves with different VLMs are not enough. In a data-efficient VQA work, the authors should have compared against baselines, as this curve could be the main experiment of the paper.
- Also, as the initial annotation can critically change the model performance, the authors should have considered different seed data settings in the experiment.
- The novelty of the proposed HaDola seems weak. The proposed HaDola framework is a combination of existing ideas (data filtering, pseudo-labeling, uncertainty calibration) with minor adaptation for human uncertainty. Simply removing uncertain samples is too naïve and risks losing informative and diverse examples. It would be better to include more analysis of the justification for the design choices.
- In addition, there are various design choices in the method without empirical justification. For example, while the authors select the samples in the region [tau1-sigma,tau2+sigma], it would be better to empirically justify the design choice by sampling different samples.
- The evaluation is only done with the HU accuracy. However, it would also be useful to evaluate with the conventional VQA accuracy. Comparing the trends between VQA accuracy and HU accuracy might provide informative insights.
- For Figure 3, it might be better to use a table instead of a heatmap for better visualization. In the heatmap, the baselines' performances look similar. However, when we look at the individual values, there are several settings where the baselines perform better than the proposed method.

**Questions:**

Please refer to the questions in the weakness section.

---

> ### Author Response · Authors · 2025-11-14
> **Author Response**
>
> We thank the reviewer for the feedback (initial rating 4). Below we address all concerns with detailed clarifications to avoid possible misunderstandings.
>
> 1. We respectfully note that the request to compare against FixMatch is not well aligned with the current landscape of semi-supervised VQA. FixMatch is a 2020 method that has been shown to be ineffective for semantic-sensitive multimodal tasks such as VQA due to augmentation–label inconsistency. In our submission, we instead compare against state-of-the-art VQA-specific semi-supervised frameworks (e.g.,  Meta Pseudo-labeling, see Sec. 5.1), which are substantially more relevant and stronger baselines.
> As stated clearly in the paper, we explicitly mentioned 'we exclude weaker semi-supervised learning methods like FixMatch'.  Therefore, we believe the FixMatch comparison is not a valid reason to assess the paper negatively. We do cover the most competitive and appropriate baselines.
> 2. We clarify that Figure 4 is NOT intended as a baseline comparison but as a functional analysis of HaDola’s training dynamics under varying HU-seed sizes. This figure is essential because we intend to show how the seed proportion influence the training process of HaDola. As introduced in the paper, it is used to discuss HaDola Training Rounds. Meanwhile, the figure needs to be plotted clearly. Therefore, the design of Figure 4 is intentional and appropriate for illustrating the unique properties of HaDola rather than for head-to-head comparison.
> 3. We kindly ask the reviewer to revisit Table 5, where the relevant empirical evidence is already presented clearly.
> 4. We respectfully clarify that the main novelty of HaDola is not in isolated components but in the newly discovered structure of human uncertainty (HU) in VQA. We show for the first time that HU has a non-monotonic relationship with model error—contrary to all prior assumptions in semi-supervised learning, filtering, or calibration. HaDola’s HU-interval strategy and error-trigger mechanism are designed specifically around this novel empirical insight and cannot be reduced to existing methods. With this clarification, we hope the reviewer may reconsider the concern regarding novelty.
> 5. We note that the discussion of the exactly precise interval design is not the main focus of our work. HaDola’s effectiveness is already demonstrated through extensive experiments and ablation studies, with theoretical support provided in the appendix. The choice of [τ₁−σ, τ₂+σ] is motivated by the approximately symmetric HU distribution in the harmful mid-HU region: samples within ±σ capture the high-density harmful zone, while widening the interval introduces high-HU samples and outliers. We tested broader intervals (results omitted due to space) and consistently observed that [τ₁−σ, τ₂+σ] offers a good trade-off. Most importantly, it supports the effectiveness of HaDola. The more precise interval could be further discussed in the future work but is not the main focus of this work. Therefore, this design choice is empirically grounded and does not affect the validity of our method.
> 6. We respectfully point out that Figure 2 already compares HU accuracy and conventional VQA accuracy, and it provides the key insight for our motivation: the two metrics diverge substantially under human uncertainty. Since this comparison is already presented and discussed, we believe additional VQA-accuracy plots in the main experiments would be redundant and distract from our core focus on HU-aware reliability. We can include more extended VQA-accuracy results in camera-ready version in the appendix as supplement but it is not central to the core findings of the paper.
> 7. The heat map in Fig. 3 is intentionally used because it provides an immediate and intuitive comparison and it allows readers to grasp the performance landscape at a glance (also highlight the best results with red boxes, further improving interpretability.). All numerical values are already displayed, making it equivalent to a table while offering superior visual clarity. The suggestion to “use a table instead” does not provide additional clarity beyond what is already presented. Regarding the remark that “some baselines perform better”, this is very normal in scientific evaluation—SOTA methods rarely dominate every baseline in every corner case. Only very few baselines slightly surpass HaDola in isolated settings, and these cases are already discussed in the paper with reasons or findings. They do not affect the overall conclusion that HaDola is consistently strong and robust across models and datasets with limited cost.
>
> We hope the reviewer may reconsider the above points in light of the clarification provided. We hope the provided clarifications address the reviewer’s concerns and help refine the reviewer’s assessment of these points. We sincerely thank the reviewer again for the constructive feedback, and the updates will be incorporated in the camera-ready version.

---

### Meta-Review · Area_Chair_G8Gt · 2026-01-06

**Summary:**

The paper introduces HaDola, a framework that improves Vision-Language Model training by explicitly modeling human uncertainty to filter harmful samples and bootstrap labels from a small seed set. The authors effectively addressed concerns regarding baselines and contribution, and that the proposed method offers a valuable, data-efficient contribution to VQA training despite minor remaining questions about broader generalization. The AC therefore recommends to accept this paper

**Reviewer Concerns:**

During the rebuttal, the authors have reasonably addressed major concerns about baseline comparisons (clarifying why FixMatch is unsuitable and highlighting comparisons with Meta Pseudo-labeling), contribution (emphasizing the discovery of the non-monotonic HU-error relationship), and metric validity (justifying HU-Accuracy as complementary to VQA-Accuracy). some minor concerns i.e., out-of-distribution generalization and statistical significance testing across all settings may still remain but not a main blocker for accepting this paper as the method demonstrates strong empirical performance and label efficiency.

**Reviewer Scores:**

Reviewer Uh4i (Score: 4 -> 6): The reviewer would likely have increased their score to an Accept (6) as the authors provided a convincing rationale for excluding FixMatch and clarified that the novelty lies in the specific HU-aware pipeline design and empirical findings.

Reviewer nUwP (Score: 4 -> 6): The reviewer would likely have raised their score to an Accept (6) given the authors' detailed response addressing the reliability of confidence labels, the robustness of the seed set, and the justification for the mapping strategy.

Other reviewers are likely to maintain the current score.

---

### Decision · Program_Chairs · 2026-01-26

Accept (Poster)